# Genome-Wide Association Study for Agronomic Traits in Gamma-Ray-Derived Mutant Kenaf (*Hibiscus cannabinus* L.)

**DOI:** 10.3390/plants13020249

**Published:** 2024-01-16

**Authors:** Woon Ji Kim, Baul Yang, Ye-jin Lee, Jae Hoon Kim, Sang Hoon Kim, Joon-Woo Ahn, Si-Yong Kang, Seong-Hoon Kim, Jaihyunk Ryu

**Affiliations:** 1Advanced Radiation Technology Institute, Korea Atomic Energy Research Institute, Jeongeup 56212, Republic of Korea; wjkim0101@kaeri.re.kr (W.J.K.); byang@kaeri.re.kr (B.Y.); yjinlee@kaeri.re.kr (Y.-j.L.); jaehun@kaeri.re.kr (J.H.K.); shkim80@kaeri.re.kr (S.H.K.); joon@kaeri.re.kr (J.-W.A.); 2Department of Horticulture, College of Industrial Sciences, Kongju National University, Yesan 32439, Republic of Korea; sykang@kongju.ac.kr; 3National Agrobiodiversity Center, National Institute of Agricultural Sciences, Rural Development Administration, Jeonju 5487, Republic of Korea; shkim0819@korea.kr

**Keywords:** kenaf, genome-wide association study, genotyping-by-sequencing, single-nucleotide polymorphism

## Abstract

Kenaf (*Hibiscus cannabinus* L.), in the Malvaceae family, is an important crop for not only fiber production, but also various other industrial materials. We performed phylogenetic analysis and a genome-wide association study (GWAS) of seven agronomic traits: days to flowering, plant height, fresh weight, dry weight, flower color, stem color, and leaf shape, using 96 kenaf genotypes, including gamma-irradiation-derived mutant lines. Genotypes were determined by genotyping-by-sequencing (GBS) and a total of 49,241 single-nucleotide polymorphisms (SNPs) were used in the analysis. Days to flowering, plant height, fresh weight, and dry weight were positively correlated with each other, and stem color was also correlated with fresh weight and dry weight. The phylogenetic analysis divided the 96 lines into nine related groups within two independent groups, and the GWAS analysis detected a total of 49 SNPs for days to flowering, plant height, fresh weight, dry weight, flower color, stem color, and leaf shape with −log_10_(*P*) ≥ 4, of which 22 were located in genic regions. The detected SNPs were located in genes with homology ranging from 45% to 96% to plants of the Malvaceae and Betulaceae, and these genes were found to be involved in plant growth and development via various pathways. Our identification of SNP markers related to agronomic traits is expected to help improve the quality of selective breeding programs for kenaf.

## 1. Introduction

A member of the Malvaceae family, kenaf (*Hibiscus cannabinus* L.), is a diploid (2n = 2X = 36) annual herbaceous plant native to Africa [1]. Originating in North Africa, kenaf is now cultivated in many countries, including India, Russia, China, and the United States, and is known to thrive in temperate and tropical environments with abundant sunlight and precipitation [1,2]. Kenaf has a variety of valuable industrial uses, including as a source of edible seed oil, stem fiber, plastic raw materials, and pharmaceuticals. In the past, the main purpose of kenaf production was to produce fiber for the manufacture of carpets, canvas bags, and rope. During World War II, the use of kenaf fiber for rope production came to the forefront, leading to research into its cultivation, production, and processing [3,4]. In Africa, kenaf leaves and stems were used to treat Guinea-worm disease and anemia, while in Ayurvedic medicine, an ancient Indian Hindu tradition, kenaf leaves were used to treat bile, blood, coughing, and diabetes [5,6]. Kenaf is known to have a crude protein content of 6% to 23% in the whole plant, and in particular, the crude protein content in the leaves reaches 14% to 34%, making it suitable as a feed source for livestock [2]. Recently, kenaf fiber has been attracting attention as a material for biodegradable, eco-friendly composites that can replace glass fiber composites [7,8]. Additionally, kenaf is an important medicinal crop as it contains high amounts of phytochemical compounds including polyphenols and anthocyanins. Thus, kenaf has been used in Indian traditional medicine (Ayurvedic medicine) as an aphrodisiac, purgative, and digestive aid including antioxidant, hepatoprotective, and anticancer activities [5].

The yield of kenaf stalks varies from 11 to 18 tons/ha, with cropping season, temperature, and soil moisture acting as major factors [2]. Kenaf plants are mainly known for their light yellow and cream-colored flowers with green stems [2]. Kenaf varieties can be categorized into three maturation groups, determined by their photosensitivity: ultra-early maturing, early-to-medium maturing, and late maturing [3]. Kenaf maturity is closely related to yield: to increase fiber yield, it is advantageous to delay flowering to increase growth rate [9]. However, if flowering is delayed too much, it is disadvantageous for seed maturation, so the photoperiod of the cultivation area according to latitude must be taken into consideration [4]. Mutation breeding involves the use of a mutagen to develop plants, exhibiting novel mutated characteristics that do not disturb elite cultivar traits [10]. Novel kenaf cultivars generated by radiation mutagenesis and showing improved seed and biomass yield characteristics have been developed using radiation breeding techniques. The cultivars of Hibiscus species have been generated via hybridization, but the floriculture and/or pharmaceutical industry relies on a limited number of mutated traits established based on specific flower quality parameters and consumer palatability [11,12,13].

Genetic change, or mutation, is a natural process that creates new genetic variants. Since the frequency of natural mutations is quite low, it is difficult to discover useful genetic variations in a short period of time. Therefore, generating genetic variations by treating crops with physical or chemical mutagens has been used successfully in mutation breeding [14,15]. Among physical mutagens, gamma rays have been used to induce mutations in various crops, such as rice, soybean, and kenaf, and to breed new cultivars [10,14,16]. Anthocyanins play an important role in plant pigmentation, which is strongly associated with the coloration in kenaf. Our research group has also developed novel kenaf cultivars via gamma irradiation on seeds of the introduced cultivars and registered them in the Korea Seed and Variety Service. In our previous work, we explored the potential of a novel flower or stem color mutation cultivar, ‘Jeokbong’ and ‘Bora’, as a functional food [14].

Detection of genomic sequences related to particular traits has the potential not only to study the function of genes, but also to accelerate the speed of future breeding [17,18,19]. Advances in next-generation sequencing (NGS) have made the rearrangement of plant genomes efficient and economical, and have also made it possible to detect the loci, types, and rates of mutations on a genome-wide scale [20,21]. GBS is a sequence-based genotyping method that is characterized by the use of restriction enzymes to sequence multiple samples simultaneously on an NGS platform using a reduced label of the target genome and a DNA barcode adapter [22,23]. GWAS is a method for detecting associations between phenotypes and genes in a population. While linkage mapping methods use bi-parental populations, GWAS is an approach that exploits diverse natural populations [24,25,26]. This approach has been applied to a variety of crops, including rice [27], soybean [28], maize [29], and sorghum [20], and it is being used in breeding programs. To date, no GWAS analyses of kenaf have been reported.

Although the potential industrial value of kenaf is quite high, genetic diversity and genetic studies of the crop are scarce. Therefore, in this study, 96 kenaf genetic resources, including gamma ray mutant lines, were sequenced using GBS, and GWAS analyses were performed for seven agronomic traits: days to flowering, dry weight, fresh weight, plant height, flower color, stem color, and leaf shape.

## 2. Results

### 2.1. Phenotypic Variation and Correlation Analysis

The 96 kenaf lines were determined for four quantitative traits: days to flowering, plant height, fresh weight, and dry weight, and three qualitative traits: flower color, stem color, and leaf shape (Table 1 and Figure 1). The days to flowering ranged from 72 to 125 d (mean 92 d) and plant height was 223–444 cm (mean 315 cm). The fresh and dry weights were 472–2400 g and 110–672 g, respectively (mean 1209 and 312 g). The coefficients of variation for days to flowering and plant height were 16.1% and 14.6%, respectively, which were significantly lower than for fresh and dry weight (43.4% and 47.9%). The skewness of all four traits was close to zero, and the height of the distribution was lower than the normal distribution. Ivory was the most common flower color (55 lines), followed by white (25) and purple (1). The most common stem color was green (89 lines), followed by dark purple (6) and brown (1). For leaf shape, there were 53 palmate types and 43 entire types. The correlations between the seven agricultural traits are shown in Table 2. The four agronomic traits, days to flowering, plant height, fresh weight, and dry weight, each had a positive correlation (*p* ≤ 0.01). In contrast, stem color showed a negative correlation with fresh and dry weight (*p* ≤ 0.05).

### 2.2. Genotyping by Sequencing of 96 Kenaf Lines

The GBS library was constructed from 96 kenaf lines, including gamma-ray-derived mutations, and sequenced using the Illumina Hiseq 2000 platform. A summary of the GBS results is presented in Table 3. Using two biological replicates, a total of 702 million reads comprising 106,096,097,764 nucleotides (106 Gb) were generated, with 7.3 million reads per genotype on average (Table 3). After trimming low quality sequences, 664,405,534 clean reads remained, with 6.6 million reads per genotype on average. The total length of the mapped region was 3,263,929,064 bp, with an average of 33,999,261 bp per sample, which covered approximately 3.17% of the whole genome; the sequences were mapped to the reference genome sequence. Among the 96 lines, the average depth of read mapping ranged from 10.84× to 21.49×.

### 2.3. Construction of Phylogenetic Tree and Genome-Wide Association Study for Agronomic Traits

Phylogenetic analysis was performed on 96 kenaf lines, including gamma-ray-derived mutants, based on the UPGMA method, and a dendrogram was generated with 49,241 filtered SNPs using the neighbor-joining method (Figure 2). In the cluster analysis, the 96 kenaf genotypes were divided into nine related groups within two independent groups, with the exception of seven lines (2012_WFM_2_3, C_11_P, C12, jangbaek_72, A19_dae, C15, and Z_1) that could not be grouped.

By plotting *r*^2^ against the distance (in kb) between a pair of SNPs, it was a very modest decrease in LD decay. When we estimated LD decay for the entire genome, the maximum *r*^2^ was halved at around 1433 kb, and as a result, the size of the LD block was considered quite large (Appendix A).

The GWAS was conducted using a multi-locus mixed model for seven agronomic traits: days to flowering, dry weight, fresh weight, plant height, flower color, stem color, and leaf shape. Using a threshold of −log_10_(*P*) ≥ 4, a total of 49 SNPs were detected across 13 chromosomes for seven agronomic traits (Table 4 and Figure 3). Of these, 12 SNPs associated with days to flowering were detected on chromosomes 1, 4, 5, 7, 10, 14, 16, and 18; 11 SNPs associated with plant height were detected on chromosomes 1, 5, 15, 16, and 18; three SNPs associated with fresh weight and dry weight were detected on chromosomes 1, 2, and 13; seven SNPs associated with flower color were detected on chromosomes 1, 2, 4, 7, and 18; five SNPs associated with stem color were detected on chromosomes 1, 11, and 17; and eight SNPs associated with leaf shape were detected on chromosomes 1, 2, 9, 11, 13, and 16. Among the 49 SNPs detected, 27 were located in intergenic regions and 22 in genic regions.

### 2.4. Gene Annotation

The gene annotations of SNPs located in genic regions detected in the GWAS analysis are shown in Table 5. The genic regions had a 45.53% to 96.34% identity with genes from the Malvaceae family (*Herrania umbratica*, *Gossypium mustelinum*, *Hibiscus syriacus*, *Durio zibethinus*, *Gossypium hirsutum*, and *Gossypium australe*) and the Betulaceae family (*Carpinus fangiana*). Except for fresh weight and dry weight, where no SNPs were detected in the genic region, the majority of SNPs were located in the exon of the respective genic regions: three out of five SNPs were detected for days to flowering, four out of five SNPs for plant height, two out of three SNPs for flower color, one out of two SNPs for stem color, and four SNPs for leaf shape. The highest level of SNPs located in an exon of a gene associated with days to flowering was found on chromosome 5 with −log_10_(*P*) = 18.86 and a 90.50% homology to the gene encoding an uncharacterized protein LOC120130459 from *Hibiscus syriacus*. The next highest −log_10_(*P*) values were found for SNPs associated with other traits: plant height showed a 64.05% homology with the gene encoding the abrin-b-like of *Durio zibethinus* on chromosome 18, flower color showed a 66.89% homology with the gene encoding the hypothetical protein FH972_000750 of *Carpinus fangiana* on chromosome 7, stem color showed an 88.18% homology with the gene encoding the pentatricopeptide repeat-containing protein of *Hibiscus syriacus* on chromosome 11, and leaf shape showed an 82.24% homology with the gene encoding the protein DETOXIFICATION 3-like of *Hibiscus syriacus* on chromosome 1.

## 3. Discussion

Kenaf has wide utility value as it can be used as a source of fiber, edible and pharmaceutical ingredients, as well as for plastic materials, but the crop’s genetics remain underresearched. The main objective of kenaf breeding has been to develop new cultivars that have high yields, are resistant to pests and diseases, are drought tolerant, or can be locally adapted to different environmental and growing conditions [5]. Breeding methods mainly used in kenaf include introduction breeding, exploiting natural variation, hybridizations, and mutation breeding. Recently, cultivars with new characteristics have become required because of changes in various cultivation environments and industrial demands [5,11]. Therefore, remote crossing and modular and mutation breeding are gradually being more widely used. However, the identification of genetic information and relationships using NGS technologies has been limited in kenaf. It is necessary to better understand the genetic basis of important traits in kenaf to improve production and to lay the foundation for molecular breeding efforts [11,12,13]. Our current study was the first to attempt GBS analysis targeting kenaf genetic resources and breeding lines for various agronomic traits and/or useful traits for the pharmaceutical industry.

In this study, we performed GWAS to investigate candidate genes for agronomic traits using 96 kenaf genotypes, including gamma-irradiation-derived mutant lines. Mutation induced using gamma radiation has been widely used to create genetic diversity via a variety of gene mutations [27]. In GWAS analyses, progeny lines can be highly correlated between samples and are liable to overestimate the genetic association because of false positives [30,31,32]. In addition, because progeny lines are derived from one or a few parent plants, they do not represent a wide range of genetic diversity, which makes it difficult to capture the many variations in the plant genome and the unusual variants that can occur via any pathway. Therefore, it is common to apply GWAS analyses to natural populations. In our results, among the lines used in the GWAS analysis, we found the phylogeny of the radiation mutant lines was sometimes close to the original line, such as the C14 lines, while in other cases, such as the Auxu and Jinju lines, they diverged into several branches (Appendix A and Figure 2). These results suggest that gamma-irradiation-derived mutant lines can be phylogenetically separated by GWAS, which uses genetic diversity wider than the original lines.

On the basis of their geographic origin, the phylogenetic analysis revealed an unclear pattern of division among the genotypes. Given that the kenaf genotypes evaluated in this study originated from various breeding methods and mixed pedigrees, it is likely that the phylogenetic analysis using many origins and races was not able to differentiate among all the genotypes. Kenaf originated in South Africa and was then introduced into India, China, Russia, and the Americas in the eighteenth century. Nowadays, kenaf is commercially cultivated in more than 20 countries, but the dissemination of genetic information about the crop worldwide is limited. Genotypes collected from Asia and central and North America were found to have close genetic relationships [33].

Correlations between agronomic traits can be used to predict yields or control harvest timing, but they are also important for breeding programs. In various crops, including kenaf, flowering time and biomass traits, such as fresh weight, dry weight, and plant height, are known to be positively correlated [34], and our results in Table 2 show the same pattern. Korean Kenaf cultivars are divided into three maturation groups depending on the flowering date: early maturing, mid–late maturing, and late maturing. Early-maturing groups mature in 70–80 d after sowing, which enables seed harvests, but at the cost of lower biomass. Late-maturing groups grow vegetatively for 130–140 d and yield significantly higher biomass, but late maturation reduces seed quality [14,15]. Therefore, a breeding goal in new Korean kenaf cultivars is to increase both biomass and seed yields per unit area [35,36]. Mutation breeding has the merits of creating new mutant characteristics and adding only few traits without disturbing the other characteristics of a cultivar [37]. The mid–late cultivars ‘Jangdae’ and ‘Wandae’, which afford both high biomass and high seed yield, have been registered with the Korea Seed and Variety Service.

Notably, there is a weak correlation between stem color and biomass. The population used in the analysis had three stem colors: green, dark purple, and brown, with stem colors other than green having a slightly lower biomass content. Although the non-green stem color was associated with lower biomass, such lines are valuable as breeding material because they produce functional substances such as anthocyanins [14]. In general, plant pigments are functional substances with various activities, such as antioxidant, antidiabetic, and anti-inflammatory compounds. Therefore, changes in stem color and flower color are accompanied by changes in the contents of functional compounds that might be valuable in various food and pharmaceutical industries. The Jeokbong, RS1, and RS2 genotypes has distinctive morphological characteristics such as dark purple color. The high levels of phenolic compounds, such as anthocyanin, observed in the previous study for the purple genotypes and their antioxidant activity are approximately 4~5 times higher than other cultivars caused by anthocyanin [14]. In the kenaf plant, the genes of phenylpropanoid biosynthetic enzymes, including *HcPAL*, *Hc4CL*, *HcC4H*, and *HcCHS*, have been isolated in previous studies [38,39]. Recent progress, including NGS technology, has facilitated various transcriptome analyses in kenaf. Previously, we successfully undertook transcriptome analysis of leaf coloration in ‘Jeokbong’, which exhibited dramatic changes in eight flavonoid structural genes. Identifying genes related to the biosynthesis of anthocyanins and kaempferitrin in kenaf leaves, 29 differentially expressed genes were assigned to eight structural genes, namely *4CL*, *CHS*, *CHI*, *F3H*, *DFR*, *ANS*, *FLS*, and *3GT* [12,14]. In addition, we performed comparative transcriptome analysis using RNA sequencing and identified putative genes (CHS, F3’H, FLS, DFR, MAT, UFGTs, TT12, GST, and RNS) involved in flower coloration at different flower developmental stages in three kenaf mutants, namely white flower, ivory flower, and purple flower [13]. When the color of petals was changed to purple or white by radiation mutation breeding, the total amount of flavonoids and phenolic compounds was more [12,13,36]. It was assumed that the accumulation of flavonoids and phenolic acids was accelerated. In the reports on the function of inside the plants, flavonoid glycosides were involved in protection against radiation stress and in producing a purple pigment [5,12,13]. We selected the SNPs in the flowers of new kenaf mutants, including white petal, purple petal, and moral types (ivory petal), as well as candidate gene data to better understand the mechanism of flower coloration via radiation.

GWAS analysis confirmed that the gene regions in which the SNPs detected for each trait were located were highly homologous to known gene regions in the Malvaceae and Betulaceae families, but no clearly identified genes for the traits were found. The LRR receptor-like serine/threonine-protein kinase (LRR-RLKs) detected as associated with flowering might be involved in several aspects of plant development, such as cell growth, differentiation, organ formation, and stress responses. Its roles might be similar to those of LRR-RLK BRASSINOSTEROID INSENSITIVE 1 (BRI1), a receptor for brassinosteroids, a class of plant hormones that regulate cell elongation and division [40,41,42]. The E3 ubiquitin-protein ligase BRE1-like 1 (HUB1), detected as associated with plant height, is known to regulate floral development in *Arabidopsis* by monoubiquitinating H2B [43]. The pentatricopeptide repeat-containing protein (PPR), detected as associated with stem color, is involved in the regulation of genes related to chlorophyll biosynthesis, photosynthesis, and flowering. It is also known to be involved in defense mechanisms against abiotic stresses, such as drought and salinity, and pathogens, such as viruses and bacteria [44]. The Korea Atomic Energy Research Institute discovered a variant with dark purple stems generated using gamma radiation as a mutation source and registered a cultivar ‘Jeokbong’ that is rich in anthocyanins (delphinidin-3-O-sambubioside). This is the same as the anthocyanin reported in roselle *(Hibiscus sabdariffa* L.). The antioxidant (2,2-diphenyl-1-picrylhydrazyl radical scavenging activities) and angiotensin-converting enzyme inhibitory activity of ‘Jeokbong’ was approximately four times higher than that of three other cultivars [14]. However, ‘Jeokbong’ and other purple stem color mutants (RS1, C14_RS1, jangjeok_S7, jangjeok_S26, and jangjeok_S25) showed poor salt tolerance. These results suggest it should be possible to develop an SNP marker for selection of stem anthocyanin and salt tolerance in kenaf.

The protein FAR1-RELATED SEQUENCE 11 (FRS11), detected as associated with leaf shape, is a member of the FRS family of transcription factors that play important roles in plant growth and development, light signaling, plant hormone responses, and stress resistance. Using a mutant with knocked out FRS11, it was confirmed that FRS11 regulates plant growth and development in response to light via its response to far-infrared light. It delayed flowering in potatoes [45] and is involved in leaf senescence in *Arabidopsis* [46]. It is clear that most of the genes identified in our study might play important roles in growth and development. However, it is likely that they influence each trait via a complex mechanism rather than a direct causal link, so further research is needed. Our findings will provide useful information for a better understanding of kenaf genetics and demonstrate the utility of mutations, either directly in breeding programs, or indirectly as a research tool.

## 4. Materials and Methods

### 4.1. Plant Materials and DNA Isolation

Ninety-six genotypes were studied (Appendix A). These included 55 mutant genotypes derived from gamma irradiation (300 Gy) of seed. Thirty genotypes were obtained from the Genebank of Rural Development Administration (RDA) in Korea and the Bangladesh Jute Mills Corporation (BJC) (Dhaka, Bangladesh). These kenaf lines were collected from South Korea (1), United States of America (1), Iran (1), India (2), Russia (2), China (7), Italy (2), and Bangladesh (14). Eight white flower mutant genotypes were generated from selection breeding (natural variants). Four genotypes were developed from hybridization between ‘Jangdae’ and ‘Baekma’ (JangXbaek21, jangXbaek21a), ‘Jeokbong’ and ‘Baekma’ (jeokXbaek66), and ‘Jinju’ and ‘Baekma’ (jinjuXbaekma). The agronomic traits investigated were days to flowering, plant height, fresh weight, dry weight, flower color, stem color, and leaf shape. Genomic DNA was isolated from 20 mg of lyophilized leaf tissue using a DNeasy Plant Mini Kit (QIAGEN, Valencia, CA, USA) according to the manufacturer’s protocol.

### 4.2. Sequence Pre-Processing and Alignment to Reference Genome

For demultiplexing, we used barcode sequences, adapter sequence trimming with cutadapt (version 1.8.3) [47], and sequence quality trimming with DynamicTrim and LengthSort programs from the SolexaQA (v.1.13) package [48]. DynamicTrim trims bad quality bases at both ends of short reads on the basis of their phred score and refines them into good-quality cleaned reads, while LengthSort removes reads with too many bases trimmed by DynamicTrim. DynamicTrim uses a phred score ≥ 20 and the LengthSort process uses a short read length ≥ 25 bp. The cleaned reads that passed the pre-processing were mapped to the reference genome using the BWA (0.7.17-r1188) program [49].

### 4.3. Raw SNP Detection and Generation of SNP Matrix

BAM-formatted files generated by mapping clean reads to the reference genome were used to perform SNP validation and raw SNP (In/Del) detection using the SAMtools (0.1.16) program [50] and SEEDERS own scripts [51], as well as to extract matching sequences. Default values were used except for the following options: minimum mapping quality for SNPs (−Q) = 30, minimum mapping quality for gaps (−q) = 15, minimum read depth (−d) = 3, maximum read depth (−D) = 398, minimum indel score for filtering nearby SNPs (−G) = 30, SNPs within INT bp around gaps to filter (−w) = 15, and window size for filtering high density SNPs (−W) = 15. We created a unified SNP matrix between samples to perform SNP comparison analysis between analytes. In this method, the raw SNP positions obtained by comparing each sample with the reference genome are used as candidate SNP positions to build a unified list, and the empty regions (non-SNP loci) are filled with the consensus sequence of the sample to create a matrix. The final SNP matrix is then created by filtering out mis-called SNP (In/Del) loci via SNP comparisons between samples. Among these loci, SNPs (In/Del) are classified according to the following type classification criteria: Homozygous, SNP read depth ≥ 90%; Heterozygous, 40% ≤ SNP read depth ≤ 60%; etc.: 20% ≤ read rate < 40% and 60% < read rate < 90%.

### 4.4. Linkage Disequilibrium Estimation

LD decay analysis with PopLDdecay program (https://github.com/BGI-shenzhen/PopLDdecay accessed on 5 December 2023) using a VCF file of 49,241 SNP loci from 96 kenaf samples. PopLDdecay Option was set as follows: Min minor allele frequency filter: 0.05, Max ratio of het allele filter: 0.6, and Max ratio of miss allele filter: 0.3. Using the calculated LD values, the LD decay plot was plotted using the HW method (the Hill and Weir method) [52] in R package.

### 4.5. Genome-Wide Association Study and Phylogenetic Analysis

To perform association analysis, we used an SNP filter process with the following conditions: SNP loci of biallelic, minor allele frequency > 5%, and missing data < 30%. The representative SNP loci and trait information selected via the filter process were used for association analysis using GAPIT [53]. The GWAS analysis was performed using a multiple mixed linear model. A total of 49,241 positions were used in the final dataset and the phylogenetic analysis was performed in MEGA6 [54]. Bootstrap consensus trees inferred from 1000 replicates were used to represent the evolutionary history of the analyzed taxa using the UPGMA method.

## 5. Conclusions

In this study, we performed phylogenetic and GWAS analyses using GBS data to investigate candidate genes for agronomic traits in 96 kenaf genotypes, including mutant lines derived from gamma irradiation. Gamma-irradiation-derived mutant lines were phylogenetically separated using wider genetic diversity compared with the original varieties. As a result of our GWAS analysis of agronomic traits, kenaf genes showing homology to known genes related to growth were discovered. The results of our study identified mutant genotypes and 19 candidate genes significantly associated with five traits of kenaf. This is the first study describing the SNPs generated using GBS and an association analysis to identify genes related to the large variability in morphological characteristics and yield-related traits among kenaf mutants derived from gamma-ray irradiation. The mutants obtained in this study will be useful genetic resources for the development of novel kenaf cultivars with improved ornamental value and chemical compound profiles. Our results may help breeders select kenaf mutants with suitable SNPs as optimal genotypes for the bioplastics and pharmaceutical industry. These findings highlight the value of mutant lines in genetic analyses and should prove useful for kenaf breeding programs.

## Figures and Tables

**Figure 1 plants-13-00249-f001:**
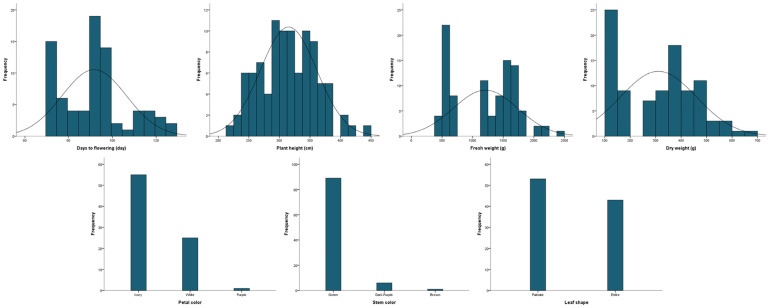
Frequency distribution of agronomic traits in 96 kenaf lines.

**Figure 2 plants-13-00249-f002:**
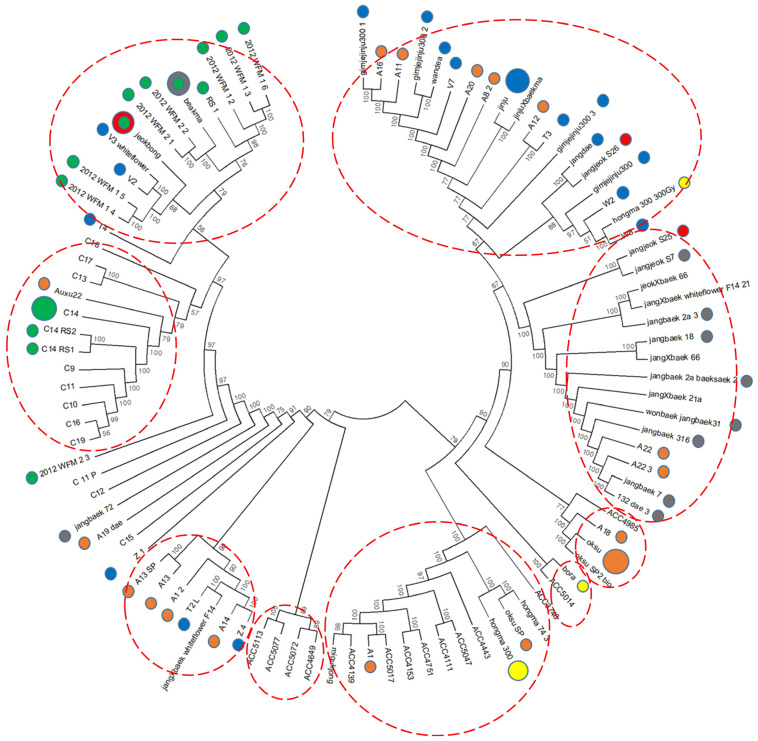
Phylogenetic tree constructed using 49,241 filtered SNP loci from 96 kenaf genotypes. Colored circles represent mutant lines; the larger colored circles denote the respective original genotypes.

**Figure 3 plants-13-00249-f003:**
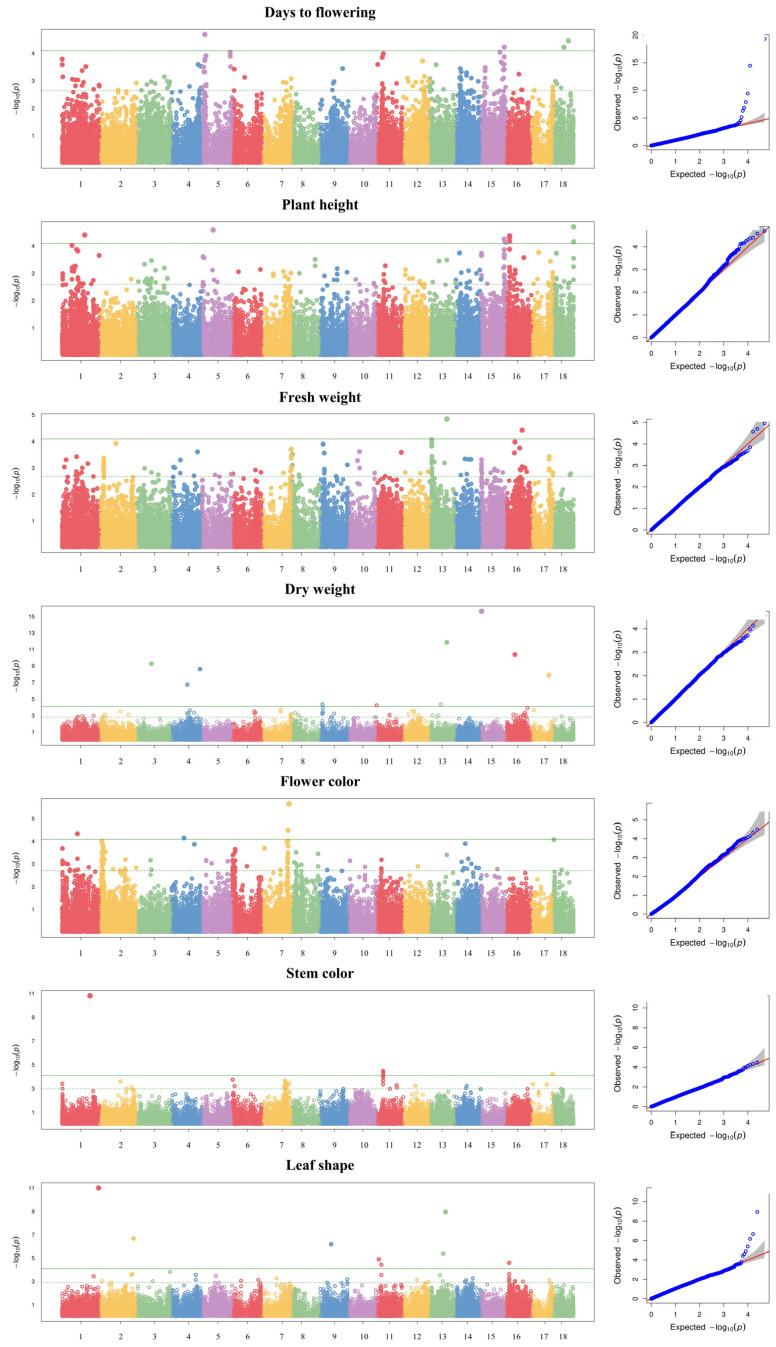
Manhattan plots and quantile–quantile plots for seven agronomic traits in 96 kenaf genotypes.

**Table 1 plants-13-00249-t001:** Descriptive statistics for agronomic traits in 96 kenaf lines.

Trait	Mean	Min	Max	SD	CV (%)	Skew	Kur
Days to flowering (d)	92	72	125	14.77	16.1	0.463	−0.410
Fresh weight (g)	1209	472	2400	525.45	43.4	−0.046	−1.211
Dry weight (g)	311	110	672	149.12	47.9	0.117	−1.043
Plant height (cm)	315	223	444	46.14	14.6	0.229	−0.309

Min, minimum; Max, maximum; SD, standard deviation; CV, coefficient of variation; Skew, skewness; Kur, Kurtosis.

**Table 2 plants-13-00249-t002:** Correlation coefficients for agronomic traits in 96 kenaf lines.

	Days to Flowering	Flower Color	Stem Color	Leaf Shape	Fresh Weight	Dry Weight
Flower color	−0.159					
Stem color	−0.128	−0.123				
Leaf shape	0.05	−0.067	−0.039			
Fresh weight	0.744 **	−0.054	−0.243 *	0.012		
Dry weight	0.725 **	−0.084	−0.229 *	−0.022	0.992 **	
Plant height	0.876 **	0.054	−0.192	−0.002	0.909 **	0.905 **

* significant at *p* ≤ 0.05, ** significant at *p* ≤ 0.01.

**Table 3 plants-13-00249-t003:** Summary of GBS sequence data and alignment to the reference genome sequence.

	Total	Mean/Genotype
Raw data		
Reads	702,623,164	7,318,991
Bases (bp)	106,096,097,764	1,105,167,685
After trimming		
Reads	635,581,208	6,620,638
Bases (bp)	63,782,931,221	664,405,534
Mapped to reference genome		
Reads	631,887,178	6,582,158
Bases (bp)	3,263,929,064	33,999,261
Reference genome coverage (%)		3.17

**Table 4 plants-13-00249-t004:** SNPs associated with seven agronomic traits detected in GWAS analysis.

Trait	Chr	Position	−log_10_(*P*)	Reference	Allele	MAF	Genic/Intergenic
Days to flowering	1	1,024,914	18.66	A	G/A	0.058	Genic
	1	42,517,384	19.30	G	A/G	0.209	Intergenic
	4	6,472,359	6.86	T	C/T	0.435	Genic
	5	6,215,497	18.86	G	A/G	0.167	Genic
	7	47,923,332	6.31	T	T/C	0.065	Genic
	10	41,332,261	4.20	T	A/T	0.310	Intergenic
	14	9,697,002	5.15	T	G/T	0.137	Genic
	14	31,710,078	14.47	C	A/C	0.069	Intergenic
	16	13,124,040	4.12	T	C/T	0.441	Intergenic
	16	23,595,487	4.51	C	T/C	0.383	Intergenic
	16	38,590,756	9.43	A	G/A	0.368	Intergenic
	18	1,013,873	7.86	A	G/A	0.291	Intergenic
Plant height	1	21,588,612	4.02	G	C/G	0.149	Intergenic
	1	48,009,505	4.40	C	T/C	0.375	Genic
	5	23,514,194	4.59	T	T/G	0.060	Intergenic
	15	48,486,135	4.25	T	A/T	0.284	Genic
	15	50,639,175	4.14	G	A/G	0.295	Genic
	15	51,634,174	4.13	G	A/G	0.327	Intergenic
	16	7,227,452	4.17	C	T/C	0.113	Genic
	16	7,227,677	4.31	A	T/A	0.122	Genic
	16	7,227,739	4.37	C	C/T	0.117	Genic
	18	42,282,127	4.70	A	G/A	0.298	Genic
	18	42,294,729	4.15	C	T/C	0.266	Intergenic
Fresh weight	1	30,956,363	4.96	A	G/A	0.218	Intergenic
	2	34,266,740	4.70	C	T/C	0.071	Intergenic
	13	36,194,770	4.57	T	G/T	0.880	Intergenic
Dry weight	1	30,956,363	4.96	A	G/A	0.218	Intergenic
	2	34,266,740	4.70	C	T/C	0.071	Intergenic
	13	36,194,770	4.57	T	G/T	0.880	Intergenic
Flower color	1	32,661,009	4.34	G	A/G	0.169	Intergenic
	2	4,123,503	4.01	T	G/T	0.426	Intergenic
	4	26,157,173	4.15	C	C/T	0.163	Intergenic
	7	54,080,566	4.02	T	C/T	0.500	Genic
	7	54,146,827	4.49	C	A/C	0.489	Genic
	7	56,094,620	5.66	A	G/A	0.240	Genic
	18	1,155,161	4.07	G	A/G	0.486	Intergenic
Stem color	1	59,001,068	10.81	G	T/G	0.069	Genic
	11	13,785,162	4.13	G	A/G	0.085	Genic
	11	13,785,316	4.47	A	A/G	0.065	Genic
	11	14,130,866	4.34	G	T/G	0.089	Intergenic
	17	44,726,748	4.23	T	A/T	0.121	Intergenic
Leaf shape	1	76,940,299	11.01	A	T/A	0.059	Genic
	2	70,676,156	6.67	A	G/A	0.161	Genic
	9	22,789,371	6.17	G	T/G	0.109	Intergenic
	11	4,977,859	4.89	C	T/C	0.307	Genic
	11	10,223,305	4.42	T	T/A	0.095	Intergenic
	13	28,565,139	5.38	C	A/C	0.104	Intergenic
	13	33,269,519	8.96	T	C/T	0.175	Intergenic
	16	6,361,215	4.59	A	G/A	0.317	Genic

Chr, chromosome; MAF, minor allele frequency.

**Table 5 plants-13-00249-t005:** SNPs and candidate genes associated with seven agronomic traits in 96 kenaf genotypes.

Trait	Chr	Position	Gene ID	Feature	Identity (%)	Location (bp)	Gene Description	Species
Days to flowering	1	1,024,914	GWHTACDB000043.1	Intron	81.36	1,021,982–1,029,723	serine/threonine-protein kinase STY46	Herrania umbratica
	4	6,472,359	GWHTACDB042287.1	Exon, CDS	45.53	6,472,165–6,473,223	hypothetical protein E1A91_D11G253500v1	Gossypium mustelinum
	5	6,215,497	GWHTACDB046286.1	Exon, CDS	90.50	6,207,495–6,233,844	uncharacterized protein LOC120130459	Hibiscus syriacus
	7	47,923,332	GWHTACDB055842.1	Intron	90.04	47,918,661–47,925,788	serine/threonine-protein kinase CTR1-like	Hibiscus syriacus
	14	9,697,002	GWHTACDB019052.1	Exon, CDS	90.14	9,690,988–9,697,964	putative LRR receptor-like serine/threonine-protein kinase	Hibiscus syriacus
Plant height	1	48,009,505	GWHTACDB002361.1	Intron	92.05	47,985,685–48,016,384	DNA polymerase I-like isoform X2	Hibiscus syriacus
	15	48,486,135	GWHTACDB023623.1	Exon, CDS	87.60	48,483,244–48,490,781	E3 ubiquitin-protein ligase BRE1-like 1	Hibiscus syriacus
	15	50,639,175	GWHTACDB023849.1	Exon, CDS	87.81	50,636,950–50,639,279	IQ domain-containing protein IQM2-like	Hibiscus syriacus
	16	7,227,452	GWHTACDB024773.1	Exon, UTR	92.59	7,227,353–7,229,462	uncharacterized protein LOC120203503	Hibiscus syriacus
	18	42,282,127	GWHTACDB032363.1	Exon, CDS	64.05	42,280,976–42,289,390	abrin-b-like	Durio zibethinus
Flower color	7	54,080,566	GWHTACDB056744.1	Exon, CDS	94.14	54,076,502–54,080,857	coatomer subunit epsilon-1	Durio zibethinus
	7	54,146,827	GWHTACDB056753.1	Exon, CDS	66.89	54,135,157–54,148,228	hypothetical protein FH972_000750	Carpinus fangiana
	7	56,094,620	GWHTACDB057049.1	Intron	94.14	56,091,648–56,099,189	PREDICTED: stomatal closure-related actin-binding protein 1-like	Gossypium hirsutum
Stem color	1	59,001,068	GWHTACDB003133.1	Intron	96.34	59,000,335–59,006,569	serine/threonine-protein kinase tricornered-like	Hibiscus syriacus
	11	13,785,162	GWHTACDB010323.1	Exon, UTR	88.18	13,784,863–13,787,951	Pentatricopeptide repeat-containing protein	Hibiscus syriacus
Leaf shape	1	76,940,299	GWHTACDB005392.1	Exon, CDS	82.24	76,939,400–76,941,687	protein DETOXIFICATION 3-like	Hibiscus syriacus
	2	70,676,156	GWHTACDB036463.1	Exon, CDS	52.79	70,673,908–70,679,842	uncharacterized protein LOC111277501	Durio zibethinus
	11	4,977,859	GWHTACDB009089.1	Exon, CDS	91.19	4,975,419–4,978,563	glutathione S-transferase DHAR3, chloroplastic-like isoform X1	Hibiscus syriacus
	16	6,361,215	GWHTACDB024720.1	Exon, UTR	94.85	6,358,025–6,361,424	protein FAR1-RELATED SEQUENCE 11	Gossypium australe

Chr, chromosome; CDS, coding sequence; UTR, untranslated region.

## Data Availability

The original contribution presented in the study are publicly available.

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
