# Peer review of "Genome-Wide Association Study for Agronomic Traits in Gamma-Ray-Derived Mutant Kenaf (Hibiscus cannabinus L.)"

_plants, 2024, doi:10.3390/plants13020249_

Round 1

Reviewer 1 Report

Comments and Suggestions for Authors

Overall, this study is important, but the structure of manuscript could be further improved. In particular, the candidate gene of leaf shape has been cloned and published in the article (Zhang et al. 2020). However, there is no information for gene annotation in Table 5. Below are some observed points that could be addressed to improve the quality of manuscript.

1.    Th number of genotypes used in this study is 96 including 55 mutant genotypes derived from gamma-irradiation (300 Gy) of several varieties. I think, the number of total genotypes is low in this study to conduct good GWAS. The minimum number of genotypes required for a genome-wide association analysis (GWAS) depending on genetic diversity of the population. Generally, a larger number of genotypes increase the power and reliability of the analysis.

2.    The title is very simple. There could be an attractive title for this study.

3.    In introduction, mention the short importance of all agronomic traits considered in this study, not just flowering time. Also, relate the reported gamma rays’ mutations in crops, e.g., kenaf with example.

4.    In Materials and Methods, the information about methodology of phenotypic data collection should be added. In addition, add the information about LD analysis and candidate genes identification. Moreover, mention the methodology of gene annotation in M&M.

5.    In results, Table Sx mentioned in line 108 but not present in the file.

6.    Why the ID of different SNPs on same chromosome is same in Table 4? Also, mention R2 value for each SNPs in Table 4 to know that how much this SNP contributes to trait phenotypic variation.

7.    Why not calculate genome wide average LD decay distance to identify the candidate genes/loci?

8.    Discussion needs improvement, try to focus on the aspects of this study.

9.    Conclusions are not comprehensive. Improve it.

Comments on the Quality of English Language

moderate editing.

Author Response

Thank you so much for your suggestion. After considering this suggestion and reviewing the manuscript, we have amended the manuscript to your suggestion.

My comments and questions to authors are as follows:

Point 1: The number of genotypes used in this study is 96 including 55 mutant genotypes derived from gamma-irradiation (300 Gy) of several varieties. I think, the number of total genotypes is low in this study to conduct good GWAS. The minimum number of genotypes required for a genome-wide association analysis (GWAS) depending on genetic diversity of the population. Generally, a larger number of genotypes increase the power and reliability of the analysis.

Response 1: As the reviewer comments, the more genetically diverse the population, the more reliable the results of a GWAS analysis. However, our genetic resources were limited, especially since we used gamma radiation to breed mutant lines and selected lines with normal growth during the breeding process, so the size of the population was limited.  As we are increasing the size of the population, future studies may show better results. Also, because this study included resources from northern, southern, and mid-latitude regions, the number of resources may be low but the genetic diversity may be high..

Point 2,3,5: The title is very simple. There could be an attractive title for this study. In introduction, mention the short importance of all agronomic traits considered in this study, not just flowering time. Also, relate the reported gamma rays’ mutations in crops, e.g., kenaf with example. In results, Table Sx mentioned in line 108 but not present in the file.                                                    

Response 2,3,5: We have revised the manuscript in response to the reviewer's comments and changed the title to "Genome-Wide Association Study for Agronomic Traits in Gamma-ray derived Mutant Kenaf (Hibiscus cannabinus L.)" and added mutant examples and Table S1..

Point 4,7: In Materials and Methods, the information about methodology of phenotypic data collection should be added. In addition, add the information about LD analysis and candidate genes identification. Why not calculate genome wide average LD decay distance to identify the candidate genes/loci?.

Response 4,7: LD analysis was performed, but since the value of 700kb was quite large and modest decrease, we decided that it may not be meaningful to find related genes in that region, so we designated candidate genes only for the region where the SNP exists, so we did not present LD separately, but according to the reviewers' comments, we will present the results of LD analysis as a supplementary figure with result.

Point 6: Why the ID of different SNPs on same chromosome is same in Table 4? Also, mention R2 value for each SNPs in Table 4 to know that how much this SNP contributes to trait phenotypic variation.

Response 6: The ids are chromosome numbers, and the GWAS analysis used a combination of ids and positions in the example below. Therefore, ids that duplicate chromosomes were removed from the table. Also, as the reviewer commented, phenotypic variation can be explained by the Rsqure value, but in the case of MLMM used in this study, it is not calculated separately because the Rsqure value of 'NA' is only present as a multiple stepes estimated method.

Point 8,9: Discussion needs improvement, try to focus on the aspects of this study. Conclusions are not comprehensive. Improve it.

Response 8,9: Based on the reviewer's comments, we have revised the introduction to the manuscript to include more information about Kenaf, as well as the Discussion and Conclusions..

Again, we appreciate the reviewer's comments.

Reviewer 2 Report

Comments and Suggestions for Authors

The article deals with the phylogenetic analysis and a genome-wide association of different agronomic traits. The article is well written and justified by several results. However, some improvements are required which are mentioned below:

In the abstract results of the “days to flowering, plant height, fresh weight, dry weight, flower color, stem color and leaf shape” should be added.

In introduction provide major producing or cultivating countries of the Kenaf.

Also add environmental and habitat requirement for its growth.

Line 63-64 lack references and should be cited with recent studies. The following studies could be added. doi: 10.1038/s41467-022-32364-3, doi: 10.1021/acs.jafc.3c02415, https://doi.org/10.1016/j.indcrop.2022.116090,

Line 71 GWAS abbreviation is already described in abstract so no need in other sections. Please be consistent. Also cite with recent studies https://doi.org/10.3390/ijms22179175, https://doi.org/10.1016/j.plaphy.2021.01.042,

What is significance of this study should be stated in the introduction.

Bast and core of the Kenaf are very important economically for fiber extraction but not characterized or studied in this study.

Why only gamma rays have been used for mutation any specific reason. There should be control treatment as well for comparison.

Figure 1 the text in the graphs are not readable.

While discussing phylogenetic analysis the author should compare the original genotypes with mutant specifically its nodes and branch length.

Figure 3 graphs have the same problem text is not readable.

In results section the phylogenetic tree should represent “into nine related groups within two independent groups” in the figure these should be distinguishing to display clear results to readers.

There are many important characterizations are missing such as intron-exon structure, subcellular localization etc. add it as limitation or provide reason or future recommendation in the discussion.

Provide links of the software and databases used in this study.

In conclusion add what is future potential of this study and what were most significant findings based on which future studies can be performed.  

Comments on the Quality of English Language

Some typos and grammatical errors are detected can be found in comments to authors 

Author Response

Thank you so much for your suggestion. After considering this suggestion and reviewing the manuscript, we have amended the manuscript to your suggestion.

My comments and questions to authors are as follows:

Point 1: Overall, the reviewer pointed out the need to improve the introduction, discussion, conclusion, add missing references, correct abbreviations, and improve the resolution of the figures. (In introduction provide major producing or cultivating countries of the Kenaf, lack references and should be cited with recent studies,  Figure text in the graphs are not readable…)

Response 1: Based on the reviewer's comments, we have revised the introduction to the manuscript to include more information about the countries and growing environments of Kenaf, as well as the Discussion and Conclusions. We have also added five references https://doi.org/10.1038/s41467-022-32364-3, https://doi.org/10.1021/acs.jafc.3c02415, https://doi.org/10.1016/j.indcrop.2022.116090, https://doi.org/10.3390/ijms22179175, and https://doi.org/10.1016/j.plaphy.2021.01.042, and retyped Figures 1 and 3 at a higher resolution.

Point 2: Bast and core of the Kenaf are very important economically for fiber extraction but not characterized or studied in this study, Why only gamma rays have been used for mutation any specific reason. There should be control treatment as well for comparison.

Response 2: As you mentioned, the fiber of Kenaf is a very important target trait, so the analysis of the quantity, physical properties, and lignin content of the fiber is underway, and this study was conducted to investigate basic agricultural traits.

In addition, the reason why we used only gamma rays is that gamma rays are mainly used as a mutagen in the institution where this researcher is affiliated, and the mutant lines used in this study were genetically fixed through generation progression over several years after mutagenesis. In addition, since we are using proton beams in addition to gamma rays to cultivate mutant strains, we believe that comparisons between mutagenic sources will be possible in future papers.

Point 3: While discussing phylogenetic analysis the author should compare the original genotypes with mutant specifically its nodes and branch length, In results section the phylogenetic tree should represent “into nine related groups within two independent groups” in the figure these should be distinguishing to display clear results to readers.

Response 3: I agree with the reviewer's comments on the phylogenetic analysis. In this study, we performed a GWAS analysis using a population containing mutant lines. GWAS analyses are typically performed using natural populations with high genetic diversity rather than progeny lines. The purpose of this study was to evaluate whether a mutant line derived from a small number of lines can have the same broad spectrum of genetic diversity as the natural population. Therefore, we did not want to evaluate the exact genetic distance between the progeny line and the mutant line, but rather the approximate distance..

Again, we appreciate the reviewer's comments.

Round 2

Reviewer 2 Report

Comments and Suggestions for Authors

Best luck for your work

Comments on the Quality of English Language

Fne